# Investigation of the *N*-Glycosylation of the SARS-CoV-2 S Protein Contained in VLPs Produced in *Nicotiana benthamiana*

**DOI:** 10.3390/molecules27165119

**Published:** 2022-08-11

**Authors:** Juliette Balieu, Jae-Wan Jung, Philippe Chan, George P. Lomonossoff, Patrice Lerouge, Muriel Bardor

**Affiliations:** 1Université de Rouen Normandie, Laboratoire GlycoMEV UR 4358, SFR Normandie Végétal FED 4277, Innovation Chimie Carnot, 76000 Rouen, France; 2Department of Biochemistry and Metabolism, John Innes Centre, Norwich Research Park, Norwich NR4 7UH, UK; 3Department of Molecular Biology, Jeonbuk National University, Jeonju 54896, Korea; 4Université de Rouen Normandie, INSERM US 51, CNRS UAR 2026, HeRacLeS, 76000 Rouen, France; 5Unité de Glycobiologie Structurale et Fonctionnelle, Université de Lille, UMR CNRS 8576, 59000 Lille, France

**Keywords:** biologics, biopharmaceuticals, vaccine, plant molecular farming, COVID, SPIKE, *Nicotiana benthamiana*, *N-*glycans, mass spectrometry

## Abstract

The emergence of the SARS-CoV-2 coronavirus pandemic in China in late 2019 led to the fast development of efficient therapeutics. Of the major structural proteins encoded by the SARS-CoV-2 genome, the SPIKE (S) protein has attracted considerable research interest because of the central role it plays in virus entry into host cells. Therefore, to date, most immunization strategies aim at inducing neutralizing antibodies against the surface viral S protein. The SARS-CoV-2 S protein is heavily glycosylated with 22 predicted *N*-glycosylation consensus sites as well as numerous mucin-type *O*-glycosylation sites. As a consequence, *O*- and *N*-glycosylations of this viral protein have received particular attention. Glycans *N*-linked to the S protein are mainly exposed at the surface and form a shield-masking specific epitope to escape the virus antigenic recognition. In this work, the *N*-glycosylation status of the S protein within virus-like particles (VLPs) produced in *Nicotiana benthamiana* (*N. benthamiana*) was investigated using a glycoproteomic approach. We show that 20 among the 22 predicted *N-*glycosylation sites are dominated by complex plant *N*-glycans and one carries oligomannoses. This suggests that the SARS-CoV-2 S protein produced in *N. benthamiana* adopts an overall 3D structure similar to that of recombinant homologues produced in mammalian cells.

## 1. Introduction

The coronavirus SARS-CoV-2 emerged in China in late 2019 and was responsible for the worldwide COVID-19 pandemic. Among the major structural proteins of SARS-CoV-2, the SPIKE (S) protein has received considerable interest because of its key role in the entry of the virus into host cells [1]. Moreover, the S protein is the most attractive SARS-CoV-2 immunogen for induction of an antibody response and is, as a consequence, the main target for vaccine and therapeutics development. The S protein is a trimeric transmembrane protein with two functional subunits. The S1 subunit binds to the cellular angiotensin-converting enzyme 2 (ACE2) of the host cell and the S2 subunit is responsible for the fusion between viral and host cell membranes. Changes in the protein S structure have been observed in the different variants of SARS-CoV-2. For example, Omicron is characterized with 37 amino acids changes in the S protein. Among these modifications, none of them concerned the *N*-glycosylation consensus sites of the S protein [1].

Because of its importance in the virus replication cycle, the *O*- and *N*-glycosylations of the S protein have received particular attention. The glycan distribution on the 22 predicted *N*-glycosylation sites (Asn-X-Ser/Thr glycosites) has been investigated either on the S1 and S2 domains or on the full-length recombinant SARS-CoV-2 S proteins produced in mammalian cells. These studies have concluded that the *N*-glycan profile is dominated by polyantennary sialylated *N*-glycans together to less abundant oligommanosides attached to specific sites [2,3,4,5,6]. With regards to the *O*-glycosylation of the S protein, 3 sites of mucin-type *O*-glycosylation have also been predicted [7] and up to 9 *O*-linked glycopeptides have been identified [2,4]. It is well documented that the glycans of the S protein are mainly exposed at its surface and form a shield masking specific epitope to escape the virus antigenic recognition. As a consequence, viral glycosylation sites are under selective pressure that may explain why SARS-CoV-2 S mutations only rarely affect amino acids of glycosites [3,6]. The presence of the glycans may also be needed for viral entry, as demonstrated by the inhibition of glycan maturation by chemical methods [8]. In addition, if the overall number of glycosites is conserved, the detailed analysis of SARS-CoV-2 S protein from coronavirus variants expressed in mammalian cells has revealed significant variations in their glycan profiles [5].

With regard to the expression of the SARS-CoV-2 S related proteins expressed in plant systems, the functional receptor-binding domain (RBD) has been successfully produced in tobacco [9,10,11,12], as well as in a plant glycosylation mutant impaired in core xylosyl- and fucosyltranferases [12] and in plants expressing glycosyltransferases responsible for the biosynthesis of human blood group A epitopes [13]. *N*-glycosylation of RBD in plants was demonstrated to be important for its proper folding. However, the alteration in the *N*-glycan processing has no impact on the protein function [12,14]. Recently, a virus-like particle (VLP) vaccine candidate bearing at its surface the SARS-CoV-2 S protein (CoVLP under the name of Covifenz) has been produced in *Nicotiana benthamiana* plants by Medicago inc. and protection against the COVID-19 provided by this VLP vaccine candidate was evaluated [15,16]. The CoVLP vaccine (Covifenz) was demonstrated to efficiently induce an immune response in humans with a 71% efficacy against various SARS-CoV-2 variants and was recently approved by Health Canada. To increase yield and stability, the version of the S protein used in these studies was modified in several ways: it contained point mutations to stabilize it in the prefusion form, the native signal sequence was replaced with a plant-derived signal sequence and both the transmembrane domain and the cytoplasmic tail were replaced with equivalent sequences from influenza haemagglutinin.

For the current study, we have chosen to use a version of the S protein that contains the native amino acid sequence, including the leader peptide evaluated [17]. This material is intended as a non-infectious surrogate for the native virus in studies on various aspects of the viral replication cycle, such as cell attachment. It is therefore important to determine how the glycosylation pattern on the plant-produced material compares with that of the native virus. Here, we report on the proteomic and glycoproteomic analysis of this plant-derived SARS-CoV-2 S protein allowing the determination of the distribution of plant *N*-glycans on the 21 *N*-glycosylation sites, as well as other post-translational modifications.

## 2. Results

### 2.1. Post-Translational Modifications of the SARS-CoV-2 S Protein Produced in N. benthamiana

The SARS-CoV-2 S protein produced in *N. benthamiana* has a molecular weight of about 140 kDa in reducing conditions [17]. Post-translational modifications of this recombinant protein were investigated by mass spectrometry analysis of peptides generated by endoprotease digestions. The plant-derived S protein was digested by different endoprotease cocktails; i.e., trypsin combined to AspN, trypsin combined to GluC or chymotrysin. The resulting peptides were analysed by nano-liquid chromatography coupled to electrospray mass spectrometry (LC-ESI MS/MS). The overall protein sequence coverage was calculated to be about 81%. With regard to termini of the S protein, the intact *C*-terminus peptide was observed (Appendix A). For the *N*-terminal end of the protein, the cleavage of the signal peptide was found to occur at Leu 10, Val 11 and Val 16. Moreover, partial oxidation of methionine residues Met 153, Met 731, Met 740, Met 900, Met 1029 and Met 1050 were detected, as well as the partial deamination of Asn 30, Asn 501 and Asn 544, pyroglutamate of Gln 675 and Gln 965, dehydratation of Thr 302 and Ser 1030, dihydroxylation of Trp 152, ammonia-loss of Asn 907, and methylation of Lys 528 and Glu 1031 (Appendix A). However, no *O*-glycosylation of Ser or Thr residues were detected, in contrast to the SARS-CoV-2 S protein produced in mammalian cells. In plants, *O*-glycosylation may also occur on hydroxyproline residues of hydroxyproline-rich glycoprotein motifs (HRGP) [18]. Moreover, the LC-ESI MS/MS analysis of SARS-CoV-2 S protein did not reveal any post-translational modifications of Pro residues.

The SARS-CoV-2 S protein is a highly *N*-glycosylated protein exhibiting 22 predicted glycosites distributed all along the protein sequence (Appendix A). Among these glycosites, only peptide containing the native Asn 1134 residue was identified in the LC-ESI MS/MS data, thus indicating that this glycosite is not substituted by *N*-glycans. For other glycosites, absence of detection of native Asn-containing peptides suggested that they were modified by the addition of *N*-glycans.

The *N*-glycosylation of the SARS-CoV-2 S protein produced in *N. benthamiana* was first investigated through the determination of its overall *N*-glycan profile. The S protein was digested by trypsin and chymotrypsin and then *N*-glycans were released from the resulting glycopeptides by PNGase A, a peptide *N*-glycosidase able to release *N*-glycans from Asn residues, even those carrying a fucose α(1,3)-linked to the proximal GlcNAc residue [19]. *N*-glycans were then purified, permethylated to improve their detection by mass spectrometry (MS) and finally analysed by MALDI-TOF MS. As illustrated in Figure 1, Gn_2_XF is the main glycan *N*-linked to the plant-derived S protein. This complex plant *N*-glycan is characterized by the presence of β(1,2)-xylose and α(1,3)-fucose residues attached to the core GlcNAc_2_Man_3_GlcNAc_2_ and arises from the Golgi processing of the oligomannose precursor synthetized in the ER [20]. In addition, biantennary *N*-glycans terminated with one or two Lewis a (Le^a^) epitopes detected (Figure 1 and Figure 2). These Le^a^ epitopes were composed of terminal GlcNAc unit, substituted by both a β(1,3)-galactose and an α(1,4)-fucose residue and derives from the late processing of *N*-linked glycans in the plant *trans* Golgi network [21].

### 2.2. N-Glycan Distribution on the SARS-CoV-2 S Protein Produced in N. benthamiana

In order to determine the distribution of *N*-glycans on the 22 *N*-glycosites of the SARS-CoV-2 S protein produced in *N. benthamiana*, a glycoproteomic approach was performed as previously reported [23]. Three independent digestions of the recombinant glycoprotein were performed by endoprotease cocktails (trypsin combined to AspN, trypsin combined to GluC or chymotrypsin). This allows the generation of multiple glycopeptides to improve the protein sequence coverage and to give better access to all glycosites along the protein. Analysis by LC-ESI MS/MS of peptides released by endoprotease digestions of the SARS-CoV-2 S protein produced in *N. benthamiana* further indicated that 21 glycosites among the 22 *N*-glycosylation sites were post-translationally modified by attachment of *N*-glycans (Appendix A). The site-specific distribution of *N*-glycans on the 22 glycosites was determined by a targeted LC-ESI MS/MS analysis of the endoprotease digests. To this end, peptides giving MS/MS spectra exhibit *N*-glycan diagnostic fragment ions at *m/z* 204 were analyzed in details.This diagnostic ion represents the oxonium ion of a HexNAc that in the context of *N*-glycan corresponds to a *N*-acetylglucosamine. A second oxonium ion at *m/z* 366 that for *N*-glycan structure corresponds to the Man-GlcNAc disaccharide was selected for in-deep MS/MS analysis (Figure 3). Using this strategy, numerous MS/MS spectra assigned to glycopeptides were extracted from data generated for the three endoprotease digestions. They all exhibited a major ion assigned to a peptide *N*-linked to a unique GlcNAc residue, thus giving easy access of the peptide mass for each selected glycopeptide (Figure 3, Appendix A). In addition to these diagnostic ions, a set of fragment ions detected by LC-ESI MS/MS allowed to rebuild the glycan sequence. As illustration, in the MS/MS spectrum of the glycopeptide V_341_-R_346_ *N*-linked to GnMXF from the SARS-CoV-2 S protein, ions at *m*/*z* 1056.52 and *m*/*z* 1553.70 revealed the presence of an α(1,3)-fucose residue on the proximal GlcNAc residue and a β(1,2)-xylose residue on the β-Man, respectively (Figure 3b). As expected, such ions corresponding to modifications of the core *N*-glycan were not observed in peptide attached to a Man-7 oligomannose (Figure 3a). In addition, a diagnostic fragment ion at *m*/*z* 512 was assigned to the Le^a^ epitope and observed in all MS/MS spectra of glycopeptides exhibiting this glycan motif (Figure 2; Figure 3c, and Figure 4; Appendix A).

Based on the fragmentation ion patterns of glycopeptides selected by the targeted LC-ESI MS/MS analysis, oligomannose, hybrid and complex plant *N*-glycans were distributed on the 21 *N*-glycosylation sites (Figure 2 and Figure 4, Appendix A). These *N*-glycans differ in the extent of their processing by Golgi resident glycosidases and glycosyltransferases during the transport of the S protein through the plant secretory pathway. These data indicated that *N*-glycan profiles differ from one site to another with mainly complex Gn_2_XF and Le^a^-containing oligosaccharides (Figure 4, Appendix A). Moreover, the number of *N*-glycans per site differs from one site to another. In contrast to other glycosites, the peptide containing the Asn-234 (Appendix A) is glycosylated by oligomannoses and hybrid oligosaccharides (Figure 4, Appendix A).

## 3. Discussion

As reported for the SARS-CoV-2 S protein manufactured in various expression systems, expression in *N. benthamiana* also resulted in the production of a heavily glycosylated protein with *N*-glycans located on 21 of the 22 glycosites. The S viral protein produced in mammalian cells exposes at its surface mature oligosaccharides including tetra-antennary polysialylated *N*-glycans [2,3,5,6]. The data reported in this study show that the tobacco-derived SARS-CoV-2 S protein is also mainly *N*-glycosylated by mature plant complex Gn_2_XF and Le^a^-containing oligosaccharides. This demonstrates that, like the S protein produced in mammalian cells, localization at the surface of the protein makes them easily accessible to glycosidases and glycosyltransferases located in the plant Golgi apparatus [20]. A similar observation was drawn for the *N*-glycosylation of the influenza haemagglutinin protein associated to virus-like particles produced in tobacco plants [24]. In contrast, the glycosite of the plant-derived S protein at Asn-234 is glycosylated by oligomannoses and hybrid oligosaccharides as reported for the viral protein produced in mammalian cells [3]. This indicates that after the protein folding in the ER, this site is buried into the protein sequence and the *N*-glycan is, as a consequence, not accessible to the Golgi processing enzymes involved in the *N*-glycan maturation.

The extent of the Golgi processing of glycans *N*-linked to a protein mainly results from its folding acquired in the ER because their maturation depends on their accessibility to Golgi enzymes as the glycoprotein moves along the secretory system. Thus, glycan maturation and distribution on a protein is indicative of its overall 3D structure. For instance, antibodies produced in mammalian cells exhibit mainly non-sialylated bi-antennary *N*-glycans. Likewise, plant-derived antibodies are *N*-glycosylated mainly by Gn_2_XF glycans and lack Le^a^ extensions. The IgG oligosaccharides are tightly associated with the CH_2_ domain of the two heavy chains which prevents their accessibility to transferases located in the *trans* Golgi apparatus [25,26,27,28]. Considering the similar distribution of oligomannoses and fully processed *N*-glycans on the 21 glycosites of the viral protein; the data on the *N*-glycosylation of the SARS-CoV-2 S protein produced in *N. benthamiana* suggest that this viral protein adopt when expressed in plants as a VLP a 3D structure similar to the one of recombinant homologues produced in mammalian and insect cells [2,3,4,5,6].

With regards to other post-translational events, no modification of the protein backbone was detected on the tobacco-derived SARS-CoV-2 S, except for the partial oxidation of some Met residues; pyroglutamate of two Gln, dehydratation of Thr and Ser, dihydroxylation of Trp, ammonia-loss of Asn, methylation of Lys and Glu, and the deamination of two Asn residues. In plants, with the exception of the *N*-glycosylation, post-translational modifications of proteins largely differ from mammalian ones with regards to the nature of structural modifications and consensus peptide sequences recognized by the enzyme machinery of the secretory pathway. Although plant is now widely used as an efficient platform for the production of human therapeutic proteins, the protein maturations performed in the plant secretory pathway may introduce non-human decorations that may impair its use in human therapy. For instance, with regards to glycans *O*-linked to plant proteins, *O*-glycosylation of hydroxyproline residues is frequently observed in plants and gives rise to hydroxyproline-rich glycoproteins (HRGP) [18]. These hydroxyprolines result from the hydroxylation of Pro residues in the plant ER. Ser residues of specific peptide motifs may also be glycosylated in plant extensines [18]. As a consequence, plant-specific *O*-glycosylation of Ser, as well as hydroxylation and *O*-glycosylation of Pro residues, have been reported on biopharmaceuticals produced in plants because peptide motifs within these recombinant protein sequence have been recognized by the plant Golgi glycosyltransferases involved in HRGP biosynthesis. This gives rise to unwanted modifications that could potentially impair the protein functionality or induce immune responses [23,29,30]. Unexpected arabinose residues have also been reported on *N*-glycans of biopharmaceuticals expressed in glyco-humanized plants. These unforseen glycan decorations likely arose from the arabinosylation of terminal galactose residues by enzymes involved in the biosynthesis of cell wall arabinan [31]. LC-ESI MS/MS analysis of the SARS-CoV-2 S protein produced in *N. benthamiana* did not reveal any post-translational modification of Ser and Pro residues resulting from enzymatic machinery involved in the HRGP biosynthesis. Taken together, the detailed analysis performed in this work indicated that the SARS-CoV-2 S protein produced in *N. benthamiana* is structurally related to homologous recombinant protein produced in mammalian and insect cells. The main difference is the structure of complex *N*-glycans, which reflects the divergence in Golgi glycosyltransferase repertoires between eukaryote cells used for the production of this viral protein. As a consequence, the plant-derived S protein exhibits non-mammalian epitopes such as core β(1,2)-xylose and α(1,3)-fucose residues. As these plant glycoepitopes are known to be putative plant allergen or immunogen motifs, immunogenicity of VLP vaccines bearing hemagglutinin proteins has been investigated. Ward et al. [15] have demonstrated that subjects enrolled in Phase I/II trials have not developed any allergy/hypersensitivity symptoms, although this vaccine protein has been reported to exposed complex and Le^a^-containing plant *N*-glycans at its surface. Altogether, the capability of plants to secrete and correctly fold large vaccine proteins makes plant-based VLP very powerful tools for the production of viral vaccines.

## 4. Materials and Methods

### 4.1. Endoprotease Digestions

VLPs containing the S protein from SARS-CoV-2 isolate Wuhan-Hu-1 (NC_045512.2) were expressed in *N. benthamiana* and purified as previously described [17]. As determined by ELISA measurement of the amount of S protein in the VLP sample, 50 µg of SARS-CoV-2 S protein were separated by electrophoresis on an 8% (*w*/*v*) SDS-PAGE gel. After staining with Coomassie Blue R250, the band corresponding to the S protein was cut into small pieces and washed several times with a 1/1 (*v*/*v*) solution of acetonitrile/100 mM ammonium bicarbonate pH 8. The S protein was then reduced with 100 mM dithiothreitol in ammonium bicarbonate pH 8 for 45 min at 56 °C and then cysteine residues were alkylated with 55 mM iodoacetamide in 100 mM ammonium bicarbonate pH 8 for 30 min at room temperature and in the dark.

For trypsin and AspN or GluC digestions, the gel pieces were first digested by trypsin (PROMEGA, Charbonnières-les-Bains, France, reference V511A in a ratio 1:20) in 100 mM ammonium bicarbonate pH 8 and placed at 4 °C for 45 min prior to incubation overnight at 37 °C. The reaction was stopped by heating to 100 °C for 10 min and the peptides and glycopeptides were recovered from the gel pieces by sequential washing with 50% acetonitrile (*v*/*v*), 5% formic acid (*v*/*v*), 100 mM NH_4_HCO_3,_ 100% acetonitrile (*v*/*v*) and finally 5% formic acid (*v*/*v*). The five elutions were combined and then dried down in a SpeedVac centrifuge (Thermo fisher, Waltham, MA, USA). The sample digested by trypsin was then treated either with AspN (ROCHE, Boulogne-Billancourt, France, reference 11054589001 in a ratio 1:50) in 200 µL of a 50 mM NaH_2_PO_4_ solution, pH 8 at 37 °C overnight. For the double trypsin/GluC digestion, the same protocol was used with GluC (PROMEGA, reference V165A) instead of AspN. As for the chymotrypsin digestion, the gel pieces were incubated with TLCK-treated chymotrypsin digestion for 3 h (SIGMA, Saint-Quentin-Fallavier, France, reference C3142, ratio 1:20) in 100 mM ammonium bicarbonate pH 8 at 37 °C. Peptides and glycopeptides were collected as described above. After digestion, the peptide and glycopeptide mixtures were heated at 100 °C for 10 min.

### 4.2. N-Glycan Profiling

50 µg of SARS-CoV-2 S protein produced in *N. benthamiana* were separated by electrophoresis on an 8% (*w*/*v*) SDS-PAGE gel. After staining with Coomassie Blue R250, the band corresponding to the S protein was cut into small pieces and washed several times with a 1/1 (*v*/*v*) solution of acetonitrile / 100 mM ammonium bicarbonate pH 8. Then, the bands of interest were submitted to successive trypsin and chymotrypsin digestions and the mixture of peptides and glycopeptides were collected as reported above. Then, *N*-glycans were released from glycopeptides with 0.02 mU of PNGase A (Roche ref 11642995001) in 50 mM sodium acetate buffer pH 5.5 at 37 °C overnight. *N*-glycans were then purified on a C18 cartridge (Thermo fisher, reference 60108-303) according to the manufacturer’s instructions, and permethylated. To this end, 1.8 g of dried sodium hydroxide was ground in 6 mL of anhydrous dimethylsulfoxyde. 0.5 mL of this mixture and 0.5 mL of iodomethane were added to the glycan sample and the mixture was incubated for 2 h at room temperature under agitation. 1 mL of HPLC grade water (Thermo Scientific, Les Ulis, France) was then added drop by drop in order to stop the reaction. One mL of chloroform and 3 mL of water were then added to the samples and mixed. The upper aqueous phase was discarded and 3 mL of water were added. This step was repeated until the pH was neutralized. The chloroform phase containing permethylated glycans was then dried down. Permethylated *N*-glycans were re-suspended in 80% methanol in water (*v*/*v*) and purified on a C18 cartridge (Thermo fisher, reference 60108-303) by successive elutions with 1.8 mL of 15%, 35%, 50% and 75% of acetonitrile in water. Permethylated *N*-glycans were recovered in the 50% acetonitrile fraction.

[M+Na]^+^ molecular ions of permethylated *N*-glycans were determined using a MALDI-TOF UltrafleXtreme mass spectrometer (Bruker Daltonics, Brème, Germany) equipped with a Nd: YAG laser (λ = 355 nm). The glycan sample was dissolved in 10 µL of acetonitrile/0.1% trifluoroacetic acid (70/30 *v*/*v*). The matrix was prepared by dissolving 20 mg of dihydrohybenzoic acid (DHB) in 1 mL of 80% methanol (*v*/*v*). 1 µL of the glycan sample and of the matrix solution were mixed and then spotted on a MALDI plate. MS Spectra were recorded in positive reflector mode an accumulation of 10,000 laser shots using a laser intensity above 80% as previously reported [32].

### 4.3. LC-ESI MS/MS Analysis

All peptide samples were resuspended in 3% acetonitrile/0.1% formic acid buffer/96.9% H_2_O (*v*/*v*/*v*). Each sample was then analyzed on a Q-Exactive Plus Mass Spectrometer coupled to an Easy nLC II system (both from Thermo Scientific, Les Ulis, France) and equipped with a nanoESI source. Peptides were loaded onto an enrichment column (C18 Pepmap100, 5 mm × 300 μm, granulometry of 5 μm, porosity of 100 Å; Thermo Scientific, Les Ulis, France) and separated on an analytical column needle (NTCC-360/100-5-153, NikkyoTechnos, Tokyo, Japan) with a flow rate of 300 μL/min. The mobile phase was composed of H_2_O/0.1% formic acid (buffer A) and acetonitrile/H_2_O/0.1% formic acid (80/20) (buffer B). The elution gradient duration was 45 min following these different steps: 0–19 min, 2–55% of B; 19–20 min, 55–100% of B; 20–30 min, 100% of B; 30–45 min, 2% of B. The temperature of the column was set at 40 °C. The mass spectrometer acquisition parameters were: 100 ms maximum injection time, 20 s dynamic exclusion time, AGC target 1 × 10^5^, intensity threshold 2 × 10^4^, 1.6 kV capillary voltage, 275 °C capillary temperature, full scan MS *m*/*z* @ 400–1800 with a resolution of 70,000 in MS and 17,500 in MS/MS. The 10 most intense ions were selected and fragmented with HCD (higher-energy collision dissociation mode) and nitrogen as a collision gas (normalized collision energy set to 27 eV). Raw data are used for subsequent spectra analysis. Protein DBsearch and PTM identification was performed by using the PEAKS studio 10.5 build 20,191,120 proteomics workbench (Bioinformatics Solutions Inc., Waterloo, ON, Canada) with the following specific parameters: enzyme, trypsin and/or chymotrypsin, AspN, GluC; 3 max missed cleavages; fixed modification, carbamidomethylation; 314 built-in variable modifications (oxidation (M), deamidation (NQ), pyro-glu from E and Q cited as example); monoisotopic mass tolerance for precursor ions, 6 ppm; mass tolerance for fragment ions, 0.02 Da; MS scan mode, FT-ICR/Orbitrap; MS/MS scan mode, linear ion trap. High confident results with a false discovery rate (FRD) below 1 only were considered. Only results presenting a −10 log P higher or equal to 30 were considered. For each glycopeptide selected by search for diagnostic ions at *m/z* 204 (*N*-acetylglucosamine) and 366 (Man-GlcNAc) as well as 512 corresponding to plant Le^a^ epitopes, manual analysis and annotation were performed for the determination of the peptide mass and glycan sequence.

## Figures and Tables

**Figure 1 molecules-27-05119-f001:**
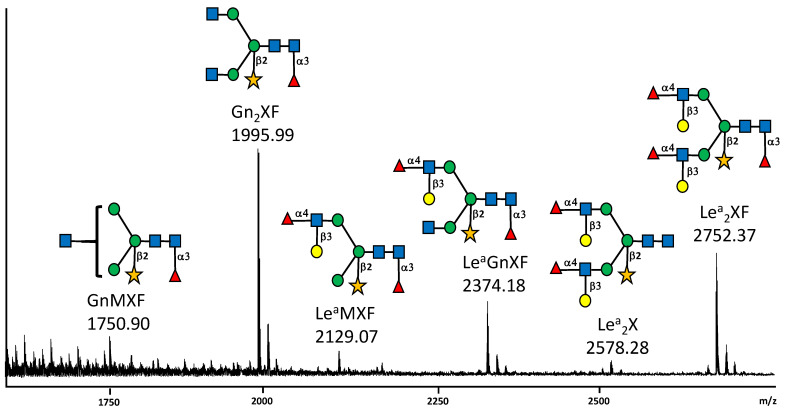
MALDI-TOF mass spectrum profile of [M+Na]^+^ molecular ions of permethylated *N*-glycans released from the SARS-CoV-2 S protein produced in *N. benthamiana*. Structures of main oligosaccharides are depicted. Each *N-*glycan has been drawn according to the international nomenclature [22]. Blue square: GlcNAc; green circle: Man; yellow circle: Gal; red triangle: Fuc and yellow star: Xyl. Linkages of monosaccharides to the core GlcNAc_2_Man_3_GlcNAc_2_ and linkage of the Le^a^ terminal antennae are indicated. See legend of Figure 2 for nomenclature of plant *N*-linked glycans.

**Figure 2 molecules-27-05119-f002:**
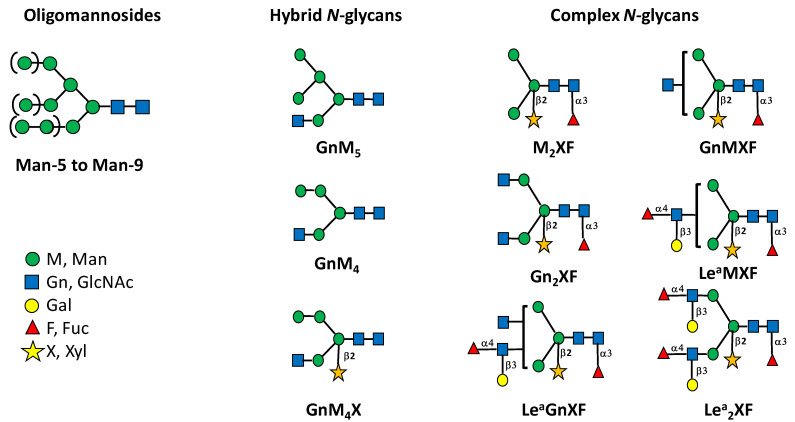
Oligomannoses, hybrid and complex glycans *N*-linked to the SARS-CoV-2 S protein produced in plants. Complex *N*-glycans are named according to the terminal glycan residues attached to Man_3_GlcNAc_2_ *N*-glycan sequence (i.e., M_2_XF stands for Man_3_(Xyl)(Fuc)GlcNAc_2_). Glycan isomers differing by their attachment of GlcNAc (Gn) or Le^a^ epitopes on the two α-mannose arms of the core are represented with a bracket (GnMXF, Le^a^MXF and Le^a^GNXF). Each *N-*glycan has been drawn according to the international nomenclature [22]. Linkages of monosaccharides to the core GlcNAc_2_Man_3_GlcNAc_2_ and of the Le^a^ antennae are indicated.

**Figure 3 molecules-27-05119-f003:**
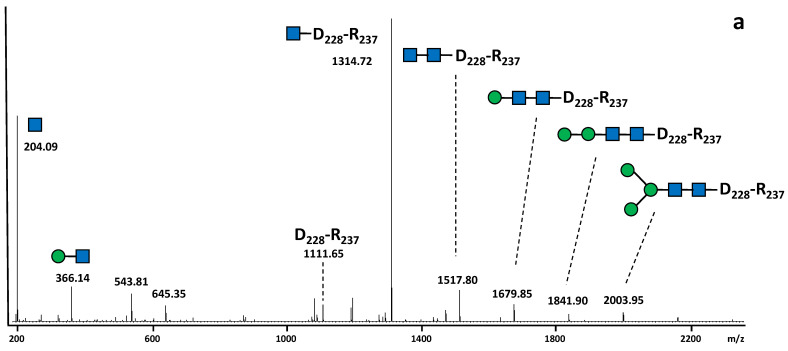
MS/MS spectra of (**a**) dicharged ion at *m*/*z* 1327.09 assigned to the peptide D_228_-R_237_ *N*-linked to Man-7, (**b**) dicharged ion at *m*/*z* 1041.44 assigned to the peptide V_341_-R_346_ *N*-linked to GnMXF and (**c**) dicharged ion at *m*/*z* 1296.54 assigned to the peptide V_341_-R_346_ *N*-linked to Le^a^GnXF. Structures of fragment ions are depicted. Each *N-*glycan has been drawn according to the international nomenclature [22]. Blue square: GlcNAc; green circle: Man; red triangle: Fuc; yellow circle: Gal and yellow star: Xyl.

**Figure 4 molecules-27-05119-f004:**
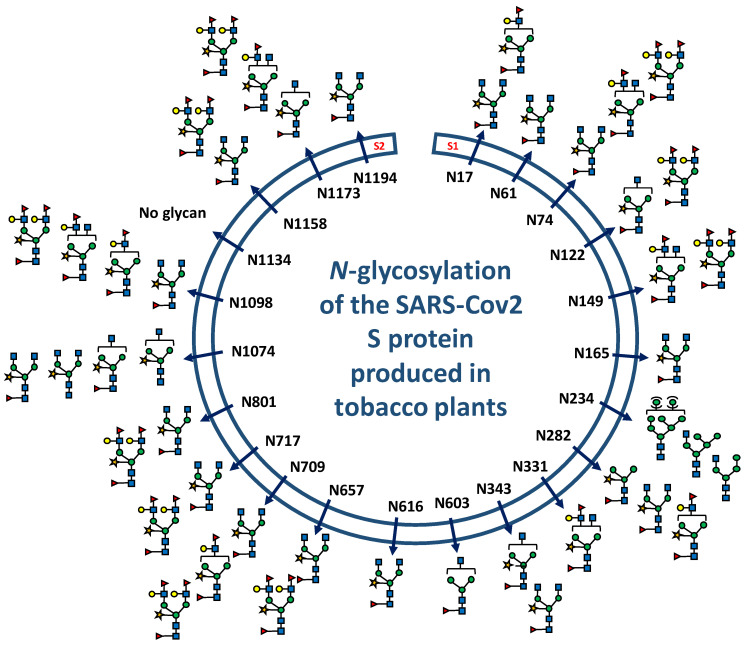
Distribution of main *N*-glycans on the 22 *N*-glycosylation consensus sites of the SARS-CoV-2 S protein produced in plants. Distribution of all *N*-glycans per glycosite are reported in Appendix A. Blue square: GlcNAc; green circle: Man; red triangle: Fuc; yellow circle: Gal and yellow star: Xyl.

## Data Availability

The mass spectrometry raw data are openly available in FigShare at doi 10.6084/m9.figshare.20209901.

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
