# Peer review of "Investigation of the N-Glycosylation of the SARS-CoV-2 S Protein Contained in VLPs Produced in Nicotiana benthamiana"

_molecules, 2022, doi:10.3390/molecules27165119_

Round 1

Reviewer 1 Report

This research provide new glycosylation data of SARS-CoV-2 S protein. The area of research is of high-impact and the data in the paper is interesting. The MS data analysis is the weak part of this paper, suggestion as follows.

Major data analysis revision/deep dive is needed for acceptance:

1) The proteomics analysis tool is too old (line 294 "Protein DBsearch was performed by using the PEAKS studio 7.5 proteomics workbench"). The related analysis do not support the conclusion (line 201 "With regards with other post-translational events, no modification of the protein bacbone was detected on the 201 tobacco-derived SARS-CoV-2 S, except for the partial oxidation of some Met residues and the deamination of two Asn 202 residues").

I would use latest proteomics open search engines such as MSFragger (https://www.nature.com/articles/nmeth.4256) or open-pfind (https://www.nature.com/articles/nbt.4236), to analysis proteomics data.

2) The glycopeptide analysis is too concise, major expansion is needed.

2.1 For each glycopeptide reported (supplementary file), what is the level of evidence the author find in the MS data? Is it only MS1 signal, or MS/MS with diagnostic ions, or high-quality MS/MS spectra (Figure 2). A supplementary table/data of detailed information for the evidence of each glycopeptide shown in Figure 4 is critical.

I do not request high-quality MS/MS spectra for each glycopeptide. Because the author has done N-glycomics analysis, diagnostic ions (204.2/366.3) and signal Y1 ion of signal should be solid evidence for glycopeptide analysis.

2.2 Semi-quantification of glycopeptides is needed in Figure4, such as XIC.

3) The reviewer can not access the RAW data from the link (line 333). Please check the link/test from a new machine to make sure the data is available.

Minor points:

4) There is significant error in Figure 2 annotation, please double check. MS spectral annotation and presentation is the core of this study.

5) From Figure 2, it seemed that the author didn't set the "first scan mass" of MS/MS spectra. For example, the lowest m/z of Figure 2B is ~120 and that of Figure 2C is ~250. Diagnostic ions in the m/z range of 120 to 200 is very useful for glycopeptide MS spectral interpretation.

If the author still has samples/MS instrument capacity, I would suggest perform the glycopeptide LC-MS data collection with the following modification: set the first mass scan value of MS/MS to 120 and using longer gradient.

Comment 5 is optional, not a requirement.

Author Response

This research provide new glycosylation data of SARS-CoV-2 S protein. The area of research is of high-impact and the data in the paper is interesting. The MS data analysis is the weak part of this paper, suggestion as follows.

Answer: we thank the reviewer for his time and valuable comments that allow us to improve the quality of the manuscript. We hope to have answer his concerns and that the paper is improved in the revised version of the manuscript.

 Major data analysis revision/deep dive is needed for acceptance:

1) The proteomics analysis tool is too old (line 294 "Protein DBsearch was performed by using the PEAKS studio 7.5 proteomics workbench"). The related analysis do not support the conclusion (line 201 "With regards with other post-translational events, no modification of the protein bacbone was detected on the 201 tobacco-derived SARS-CoV-2 S, except for the partial oxidation of some Met residues and the deamination of two Asn 202 residues").

I would use latest proteomics open search engines such as MSFragger (https://www.nature.com/articles/nmeth.4256) or open-pfind (https://www.nature.com/articles/nbt.4236), to analysis proteomics data.

 Answer: As per the suggestion of the reviewer, we tried to use both MSFragger and open-pfind to reanalyse the data and compare these additional results to the one previously obtained with the PEAKS 7.5 software. Unfortunately, neither MSFragger or pfind allow us to obtain results while running as they were error message impacting some of the tools fonctions. These could not be fixed during the revision timeframe even contacting the support team of both tools. As a consequence, in order to take into account the valuable reviewer comment, we uploaded an updated version of the PEAKS softare, namely the version 10.5 from 2019 and re-run the analysis. More PTM have been identified, they are listing in the text of the revised version of the manuscript and supported by an additional Figure S2 in the supplementary material. The experimental procedure has been revised accordingly.

2) The glycopeptide analysis is too concise, major expansion is needed.

Answer: Accordingly to this comment, we expanded the section regarding the glycopeptide analysis in the revised version of the manuscript. We hope it reads better now. Moreover, for an easier understanding of the reader, we added a supplemental Figure 1 that contained the amino acid sequence of the recombinant S protein expressed in tobacco plants.

2.1 For each glycopeptide reported (supplementary file), what is the level of evidence the author find in the MS data? Is it only MS1 signal, or MS/MS with diagnostic ions, or high-quality MS/MS spectra (Figure 2). A supplementary table/data of detailed information for the evidence of each glycopeptide shown in Figure 4 is critical. I do not request high-quality MS/MS spectra for each glycopeptide. Because the author has done N-glycomics analysis, diagnostic ions (204.2/366.3) and signal Y1 ion of signal should be solid evidence for glycopeptide analysis.

Answer: We completed the Supplemental Table I file with the MS/MS diagnostic ions detected for each glycopeptide as well as the information regarding the ion corresponding to the glycopeptide containing a GlcNAc residue from the core of the N-glycans.

2.2 Semi-quantification of glycopeptides is needed in Figure4, such as XIC.

 Answer: Such information has been extracted as per the request and the relative quantification for each glycopeptide expressed as a % for each N-glycosylation site is now including in the new supplemental Table S1 of the revised version of the manuscript.

3) The reviewer can not access the RAW data from the link (line 333). Please check the link/test from a new machine to make sure the data is available.

Answer: We thank the reviewer for this comment. We did check the link and a DOI has been asigned to the dataset that is now  10.6084/m9.figshare.20209901. This information has been modified in the manuscript. As per the rules of FigShare, the DOI becomes active when the item is published.

Minor points:

4) There is significant error in Figure 2 annotation, please double check. MS spectral annotation and presentation is the core of this study.

Answer: The figure 2 has been checked carefully and few mistakes have been corrected in the revised version.

5) From Figure 2, it seemed that the author didn't set the "first scan mass" of MS/MS spectra. For example, the lowest m/z of Figure 2B is ~120 and that of Figure 2C is ~250. Diagnostic ions in the m/z range of 120 to 200 is very useful for glycopeptide MS spectral interpretation.

If the author still has samples/MS instrument capacity, I would suggest perform the glycopeptide LC-MS data collection with the following modification: set the first mass scan value of MS/MS to 120 and using longer gradient.

Comment 5 is optional, not a requirement.

Answer: We thank the reviewer for highlighting this. The Figure 2C has been changed and is now starting earlier than 250. As for the suggestion of running a longer gradient, we feel that with the Qexactive instrument, it will not bring any benefit. Indeed, the scan speed used in the mass spectrometer is suffisant enough to obtain high resolution data with the gradient employed. Previous attempt to increase the lenght of the gradient did not improve the resultats and in contrast leads to lost of informations due to the lower resolution of each individual chromatography peak that was broaden.

Reviewer 2 Report

The manuscript described the Investigation of the N-glycosylation of the SARS-CoV-2 S 3 protein contained in VLPs produced in N. benthamiana the paper is well written the following should be considered

Authors should compare the changes in the omicron virus and other variant of SARS-CoV-2 S 3 proteins at a suitable places in the manuscript with proper references the authors may go through this recent article https://doi.org/10.1002/jmv.27936  or any other suitable references.

Author Response

Answer : We thank the reviewer for this suggestion and valuable comment. Based on the comment and in order to make it clear for the reader regarding which sequence has been expressed in tobacco plants, the amino acids sequence of the expressed protein has been added in supplementary material as Figure S1 in the revised version of the manuscript. Moreover, changes observed in the omicron and other variants have been highlighted in the introduction of the revised version. As per the suggestion of the reviewer, the recent article has now been cited.

Round 2

Reviewer 1 Report

The authors have addressed all concerns in MS data interpretation and glycopeptide data presentation. Current version is suitable for publication.